# Acceleration Skinning: Kinematics-Driven Cartoon Effects for Articulated Characters

Niranjan Kalyanasundaram *
Clemson University

Damien Rohmer†
LIX, École Polytechnique/CNRS, IP Paris

Victor Zordan‡
Clemson University
Roblox

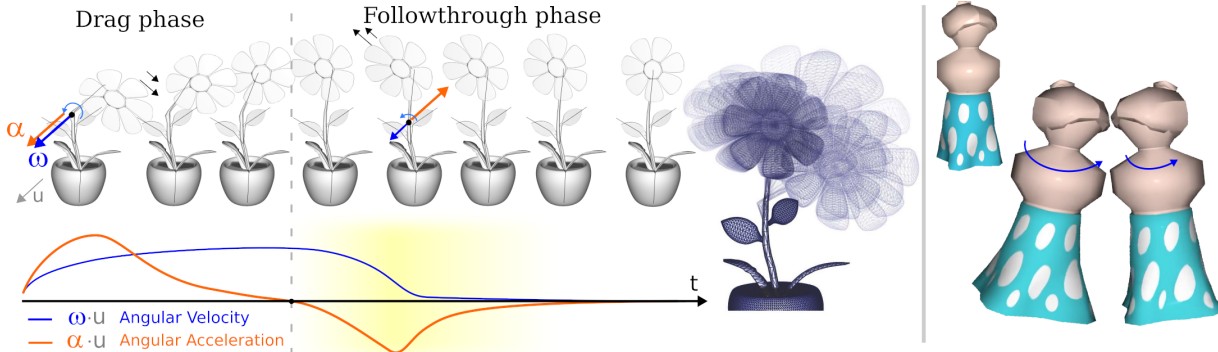

Figure 1: Real-time cartoon-like deformation computed automatically with *Acceleration Skinning* extending the range of deformation available from velocity only. Left: The joint use of angular velocity $\omega$ and acceleration $\alpha$ allows a smooth variation between an acceleration phase (where $\omega$ and $\alpha$ have the same orientation) associated with a drag effect, and a deceleration phase (where $\omega$ and $\alpha$ have opposite signs) associated with a "followthrough" transition. Right: A *centrifugal effect* coupled with a *lift deformer* creates the lift of the rotating skirt of this dancing toy character.

## ABSTRACT

Cartoon effects described in animation principles are key to adding fluidity and style to animated characters. This paper extends the existing framework of *Velocity Skinning* to use skeletal acceleration, in addition to velocity, for cartoon-style effects on rigged characters. This *Acceleration Skinning* is able to produce a variety of cartoon effects from highly efficient closed-form deformers while remaining compatible with standard production pipelines for rigged characters. The paper showcases the introduction of the framework along with providing applications through three new deformers. Specifically, a *followthrough* effect is obtained from the combination of skeletal acceleration and velocity. Also, *centrifugal stretch* and *centrifugal lift* effects are introduced using rotational acceleration to model radial stretching and lifting. The paper also explores the application of effect-specific time filtering when combining deformations together allowing for more stylization and artist control over the results.

**Index Terms:** Computer Animation, Procedural Deformation.

## 1 INTRODUCTION

The fundamental principles of animation laid out by Walt Disney Studios remain an important driving factor for animators today [26]. These principles, created by the pioneers of traditional hand-drawn animation, have been guiding artists on how to create effects such as *squash and stretch* and *followthrough* to add vitality to animation. Many techniques have been proposed to adapt such cartoon effects to suit the world of 3D animation production [14], however their application still remains a non-trivial task. Currently, professional

---

*e-mail: niranjan@nkportfolio.com
†e-mail: damien.rohmer@polytechnique.edu
‡e-mail: vbz@clemson.edu

animators generally create such effects either through rigged animation or using custom, manually controlled deformers [17]. While these approaches allow fine artistic control, they become tedious in adapting the work to new characters and animated sequences. As an alternative, physically-based simulation can produce the effect in a fully automatic fashion, but often requires expert tuning and costly computation which does not adapt easily to the usual production pipeline, including the iterative animation workflow.

Recently, a technique called *Velocity Skinning* [23] (VS) introduced a method able to automatically represent a subset of cartoon effects, including *squash and stretch* and *drag/floppy* effect, that can be applied on top of a character deformed by skinning solely using a standard rig. As its name suggests, it relies on the use of velocity at the joint level to drive the deformations, providing real-time effects that automatically adapt to the animation and hierarchy of the rigged character. The approach provides simple and direct artist control over the deformation that works with animator workflow, and does not require costly physical simulation.

In this paper, we propose a natural extension we call *Acceleration Skinning* (AS) which builds off of this previous work. Namely, we explore and showcase additional effects for skinning animation based on the use of acceleration in the pipeline, in conjunction with the VS approach. To this end, we propose a more general framework that combines both, as described in Sec. 3. This generalization allows a wider set of deformers while preserving the original advantages of the VS approach, such as its high efficiency and art directability. Foremost, we showcase the additional use of acceleration, in combination with velocity, to model overshooting deformation, a common effect which is often called *followthrough* in animation guidebooks (Sec. 4). Further, as acceleration provides a natural separation of coordinate frames, e.g. associated to centrifugal effects when rotational motions are involved, we also propose additional parameterized deformation effects. Specifically, we introduce *centrifugal stretch*, for the elongation of an object that rotates, and *centrifugal lift*, for the elevation of a rotating object, such as a twirling skirt (Sec. 5). Finally, we extend the general (VS/AS)

domain further by exploring the use of time filters to highlight and tune effects based on artist control (Sec. 6).

In total, the contributions of this paper are to extend the capabilities of VS and skinning overall, and add a set of interesting, AS-specific effects with little overhead in implementation and cost while still supporting a convenient workflow for artist-driven cartoon animation.

## 2 RELATED WORK

Much of the production animation created today for characters depends on skinning, such as Linear Blend Skinning (LBS) [16], where the surface geometry (skin) of the character is moved in relation to an articulated skeleton. The dependence between the joints of the skeleton and the vertices of the skin is defined by scalar *skinning weights* and in combination with the skeleton and joints constitute the *rig* for character skinning. LBS is a straightforward implementation of skinning where the deformation is computed as a linear combination of the joint transforms. A large set of work in computer animation research has improved upon LBS skinning techniques but LBS remains a common tool in practice. As such, our general formulation for *Acceleration Skinning* is derived for the basic LBS formulation, but the acceleration-skinning deformers proposed may also be applied to other methods, such as the *Dual Quaternion* approach [8], another common skinning technique.

Skinning techniques define (static) skin deformation of the character. This deformation is guided from a skeletal pose traditionally derived from interpolating keyframes. However, this approach only interpolates in-between poses and does not model dynamic effects, such as drag at high speeds, or followthrough due to inertia. To this end, physically-based simulation has been used to represent so-called *secondary actions* on top of keyframe motion. We distinguish three main approaches to compute secondary motion. First, projection-based dynamics [18] and its extensions [3] have been proposed to modeling muscle and soft tissue giggling effects [10,19,25] and can be tuned to exaggerate specific cartoon-like effects [2,4]. Second, layering techniques have been proposed where distinct effects are handled as simple physical models, using jiggling implicit surfaces [22], spring-driven bones [11], or custom volume effects such as squash and stretch [1]. Third, reduced deformable models have been introduced [6,28,30] where precomputed shape analysis allows for model reduction, and therefore sped-up computation. Related techniques for producing simulation effects are still being proposed as well, such as efficient simulated cartoon deformations applied on top of skinning animation [31], which defines the space of secondary effects as the orthogonal subspace of the rig. In total, these physically-based approaches are able to model rich and detailed dynamic exaggerated deformations, possibly with collision handling, but they share two main drawbacks. First, physically based simulations are limited with respect to efficiency. Even fast approaches, that can tolerate models with thousands of vertices, cannot scale easily to a large number of meshes at the same time –as may be required for video games applications– or very detailed mesh of several millions of vertices, while geometric methods (such as VS and AS as well as traditional skinning) are still orders of magnitude faster. Second, simulation requires numerical time integration computed from the initial conditions. This integration imposes the animation to be *baked* before being visualized, which hampers the efficiency of the animator by limiting the possibility of navigating and editing details at arbitrary instances along the animation timeline. As such, standard production pipelines [17,21] often favor procedural deformers that are independent of past state when possible.

Aside from simulation, geometric and kinematics-based deformations have also been proposed in the literature. For instance, sketch- [9,15] and example-based [5,24] techniques aim to precisely define shape deformations for expressive effects, but require manual input for each shape. In contrast, procedural approaches allow more automatic adaptation while remaining compute efficient, as they take advantage of the skeletal structure position and kinematics. For example, Nobel et al. [20] introduce a local bending behavior parameterized by the direction and velocity between consecutive bones. Yong et al. [12] propose a *squash and stretch* deformer parameterized by root joint velocity, and a *drag* effect on end-effector joints of a skinned character. Procedural bone stretching was also proposed by Kwon and Lee [13] for local *squash and stretch* effects. Recently, Rohmer et al. [23] introduced *Velocity Skinning* (VS) that can be seen as a generic framework to define local deformers such as *squash and stretch* and *drag* based on joint velocity. Like the proposed work here, the VS approach fits well to standard animation pipeline in reusing the existing rig (i.e. original skeleton and skinning weights) for automatic parameterization of deformation. VS also provides a closed form procedural deformation that can be computed extremely efficiently for each vertex in parallel, similar to basic skinning. Our approach follows VS, but extends the approach to add acceleration terms. This allows additional deformation behaviors that cannot be achieved in using velocity alone such as the proposed *followthrough* effect, that requires deformation as the velocity approaches zero. We also note some cartoon effects have been generated using time filtering [27] or oscillating splines [7] applied to vertex trajectories. Our approach also parallels these works in applying low-pass time filters in order to control the effect-specific timing and magnitude of effects relative to the instantaneous joint values of the skeleton.

## 3 ACCELERATION SKINNING

In LBS, the deformed position $p$ of an arbitrary vertex at an initial position $p^0$ can be expressed as the linear blending

$$p = \sum_j b_j \, p_j \,, \tag{1}$$

where $j$ is the index of a joint, $b_j \in [0,1]$ are the skinning weights that should satisfy $\sum_j b_j = 1$. $p_j$ are the *rigid skinning* deformation of the position $p^0$ relative to the joint $j$. As shown in Rohmer et al [23], differentiating this expression in time, and reorganizing the terms along the skeleton hierarchy leads to the following "skinning-like" relation in velocity

$$v = \sum_j \tilde{b}_j \, v_{/j} \,, \tag{2}$$

where $v$ is the net velocity of the vertex at position $p$, $v_{/j}$ is the contribution coming from the rigid transformation of the joint $j$ only to the net velocity. And $\tilde{b}_j$ are the *kinematics-weights* that are obtained from a combination of skinning weights

$$\tilde{b}_j = \sum_{k \in Desc(j)} b_k \,, \tag{3}$$

with $Desc(j)$ being the descendants of the joint $j$ in the skeleton hierarchy.

We note that this derivation is valid for any arbitrary degree of differentiation in time. Therefore, relation in Eq. (2) can also be differentiated, which leads to an acceleration-like skinning

$$a = \sum_j \tilde{b}_j \, a_{/j} \,, \tag{4}$$

with $a$ being the net acceleration of the vertex at position $p$, and $a_{/j}$ is the contribution coming from the joint $j$ to the net acceleration.

Both velocity $v_{/j}$ and acceleration $a_{/j}$ contributions are related to rigid motions, and we can therefore breakdown their expressions in relation to linear/angular velocity and acceleration of their respective joint as

$$\begin{aligned} v_{/j} &= v_{Lj} + \omega_j \times r_j \\ a_{/j} &= a_{Lj} + \alpha_j \times r_j + \omega_j \times (\omega_j \times r_j) \,. \end{aligned} \tag{5}$$

$v_{Lj}$ and $a_{Lj}$ are respectively the linear velocity and acceleration of joint $j$. $\omega_j$ is the angular velocity, and $\alpha_j = \dot{\omega}_j$ is the angular acceleration vector. $r_j = p - \text{proj}_j(p)$ is the relative vector between $p$ and its orthogonal projection $\text{proj}_j(p)$ onto the rotation axis passing by the joint $j$ and oriented along $\omega_j$. Note that while velocity depends only on two components (linear and angular), the acceleration depends on three, respectively, the linear component ($a_{Lj}$), *angular* component ($\alpha_j \times r_j$), and *centripetal* component $\omega_j \times (\omega_j \times r_j)$. These components are illustrated in Figure 2 for velocity and acceleration.

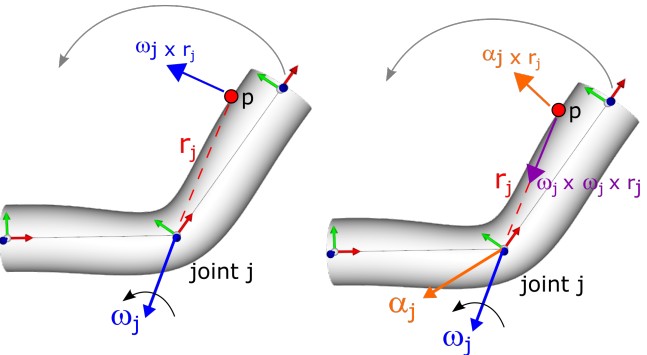

Figure 2: Angular velocity (left) and acceleration (right) components associated to the circular motion of joint $j$. The acceleration can be split between the angular component (orthogonal to $\alpha_j$ and $r_j$) and a centripetal one (oriented toward the rotation axis directed by $\omega_j$).

Aligned with Rohmer et al. [23] the general idea in acceleration skinning is to define "deformers" ($\psi$s) as procedural functions representing a specific type of parameterized deformation at the individual joint level. Subsequently, the deformers are combined over the skeleton hierarchy as Eq. (4) in order to distribute globally the total deformation over the skinning surface as

$$d(p) = \sum_j \tilde{b}_j \, \psi(v_{Lj}, a_{Lj}, \omega_j, \alpha_j, r_j) \,. \qquad (6)$$

$d(p)$ is the net deformation to $p$ such that the final deformed position is $p_{final} = p + d(p)$. Further, the deformer $\psi$ can itself be decomposed into a linear sum of individual sub-effects $\psi^{\text{effect}}$ such that

$$\psi = \sum_{\text{effects}} \psi^{\text{effect}} \,. \qquad (7)$$

In this paper, we adapt the deformers from the Velocity Skinning work (e.g. drag) and propose three unique deformers that are constructed using the components of the derived acceleration terms from Eq. 5. Namely, we propose formulations for new deformers that we call respectively $\psi^{ft}$ for followthrough, $\psi^{cs}$ for centrifugal stretch, and $\psi^{cl}$ for centrifugal lift. These are described in the following sections in detail. As each deformer $\psi^{\text{effect}}$ is defined in a generic way for any joint $j$ we will omit to explicitly mention the dependence to the index $j$ in its velocity and acceleration parameters for notation clarity.

## 4 FOLLOWTHROUGH DEFORMER

*Followthrough* is a fundamental principle of animation and has been used extensively in animation throughout history as a way to convey the inertia of the animated shape (see Figure 3). Drag and followthrough are two effects that often go hand in hand. We describe our approach to represent followthrough effect next, relying on the combination of velocity and acceleration information to extract the salient details needed to construct the effect.

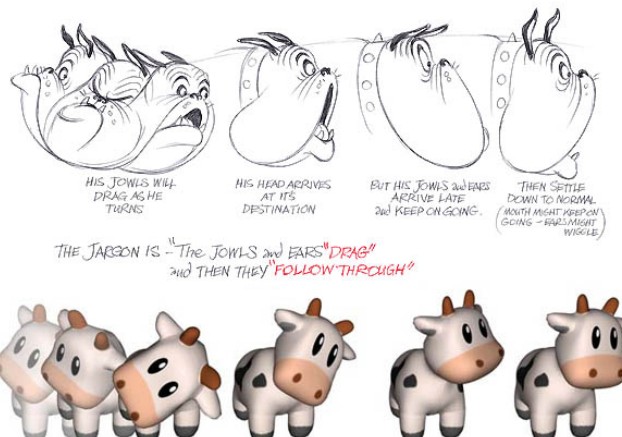

Figure 3: Top: Concept of drag and followthrough explained. Image courtesy of The Animator's Survival Kit [29]. Bottom: Acceleration Skinning animation results utilizing proposed acceleration-based drag and followthrough deformers.

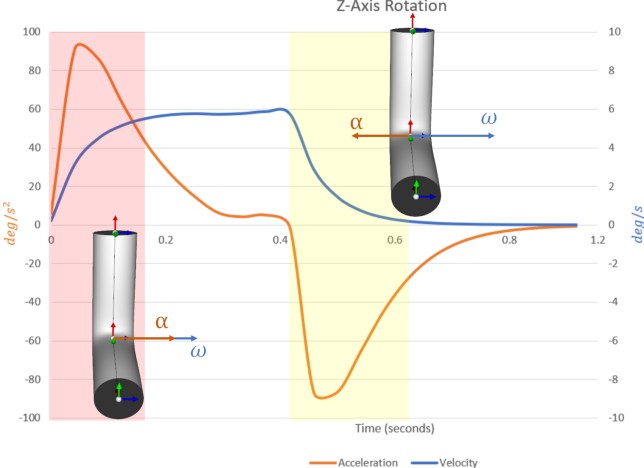

Figure 4: Three phases during a circular motion illustrated by their angular velocity $\omega$ and acceleration $\alpha$ curve along time. Left (red): acceleration phase where $\alpha$ and $\omega$ are aligned and oriented in the same direction. Middle: Constant velocity magnitude with near zero angular acceleration. Right (yellow): deceleration phase where $\alpha$ and $\omega$ are opposed.

A velocity-based *floppy* deformer $\psi^{floppy}$ was proposed in VS to model a drag effect. This deformer was split into two parts either coming from the linear velocity or the angular velocity. The linear velocity related effect was associated with the deformer $\psi^{floppy,lin}$ representing a translation, while the angular velocity related effect was associated with the deformer $\psi^{floppy,ang}$ representing bending.

$$\begin{aligned} \psi^{floppy,lin}(v_L) &= -K_{lin} \, v_L \\ \psi^{floppy,ang}(\omega, r) &= (R(-K_{ang}\|\omega \times r\|, \omega) - I) \, r \,, \end{aligned} \qquad (8)$$

where $R(\theta, u)$ is the rotation matrix parameterized by an angle $\theta$ and an axis $u$, and $I$ is the identity matrix. $K_{lin}$ and $K_{ang}$ are coefficients that an animator may use to scale the magnitude of each deformation separately.

A straightforward extension of such to model followthrough consists of substituting the linear and angular acceleration terms instead

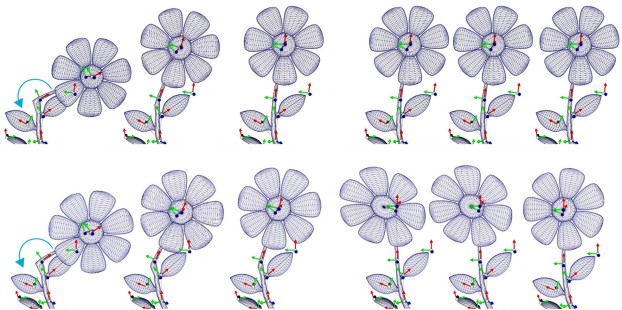

Figure 5: Flower model with circular motion. Top: Source LBS deformation, Bottom: After application of *followthrough* deformer $\psi^{ft}$. The deformation is visible acting at the fourth and fifth image in the sequence.

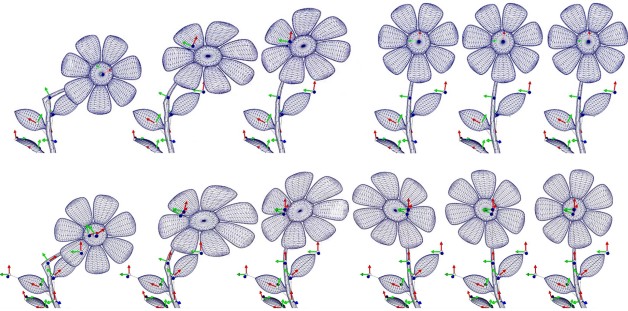

Figure 6: Top: application of the acceleration drag $\psi^{ad}$. Bottom: Mix of acceleration drag $\psi^{ad}$ with followthrough $\psi^{ft}$. Note as the magnitude of these rotation-based deformers varies with the distance between the vertices and the center of rotation, the petals appears to squash and stretch locally.

of their velocity counterpart in these floppy deformers. However, this substitution leads to undesirable artifacts. Consider a generic motion (linear or angular) with three phases: an acceleration phase at its start; a constant velocity phase; and a deceleration phase at the end before the motion stops. These three phases are illustrated for a simple angular motion in Figure 4 (and Figure 1, left) comparing the velocity-based deformer to the acceleration-based one. For such a case, the proposed acceleration-based deformer will first start by applying a draglike deformation, continue with no deformation, and end during the deformation phase with an opposite effect of drag, i.e. an overshoot of the deformation beyond the final pose.

This last phase corresponds to the expected behavior for followthrough representation. However, the proposed deformer also contains a draglike behavior followed by no deformation which is not expected. To avoid this unexpected behavior, we modified the proposed deformer by filtering based on the relative direction between the current velocity and acceleration parameters. More precisely, we introduce a smooth indicator function $\mathscr{D}$ that we characterize by the condition that the velocity component is aligned and opposite with the acceleration component. Specifically, we propose the following

$$\mathscr{D} : (a,b) \mapsto \begin{cases} 0 & \text{if } a \cdot b \geq 0 \\ |a \cdot b|/D & \text{if } 0 \geq a \cdot b \geq -D \\ 1 & \text{otherwise} \end{cases} \qquad (9)$$

where the parameter $D \in [0,1]$ allows to adapt how fast the transition to the deceleration phase is computed. We then define the followthrough (ft) deformer, $\psi^{ft}$, as

$$\begin{aligned} \psi^{ft,lin}(v_L, a_L) &= -\mathscr{D}(v_L, a_L) \, K_{lin} \, a_L \\ \psi^{ft,ang}(\omega, \alpha, r) &= \mathscr{D}(\omega, \alpha) \, (R(-K_{ang}\|\alpha \times r\|, \alpha) - I) \, r \, . \end{aligned} \qquad (10)$$

Results obtained using this deformer are illustrated on a circular motion in Figure 5.

In addition, we can further take advantage of the smooth indicator to separate the contribution of drag from the followthrough. Doing so, we propose additional pure acceleration drag (ad) deformer $\psi^{ad}$ as

$$\begin{aligned} \psi^{ad,lin}(v_L, a_L) &= -(1 - \mathscr{D}(v_L, a_L)) \, K_{lin} \, a_L \\ \psi^{ad,ang}(\omega, \alpha, r) &= (1 - \mathscr{D}(\omega, \alpha)) \, (R(-K_{ang}\|\alpha \times r\|, \alpha) - I) \, r \end{aligned} \qquad (11)$$

which will apply a drag effect during the *acceleration* phase of the motion (see Figure 6, top). Note that both deformers $\psi^{ft}$ and $\psi^{ad}$ can be used together on an animation and parameterized separately in adapting their respective coefficients $K_{lin}$ and $K_{ang}$ for additional

user control (see example in Figure 6,bottom). Note that this notion of combining drag with followthrough remains consistent with traditional animation concepts [29] (Figure 3, top).

## 5 CENTRIFUGAL-BASED DEFORMERS

Centrifugal motion during rotation of a physical system moves away radially from the axis. This movement can be countered by centripetal acceleration if conditions are favorable. However, we can also exploit this relationship to propose new deformation effects. Namely, as noted in Eq. (5), the net acceleration of a vertex following a rotating motion contains a *centripetal* component acting radially around the rotation axis, and expressed as $\omega \times (\omega \times r)$. This component exists even when the magnitude of the velocity remains constant, as in the case of a circular motion with constant angular velocity. We employ this acceleration in two distinct effects, first, with a generic centrifugal stretch deformer $\psi^{cs}$, and second, in a centrifugal "lift" deformer $\psi^{cl}$. By their nature, both are applicable for rotating motion only.

### 5.1 Centrifugal stretch

When observing how chefs make pizza, we see that after kneading and pressing the dough, they proceed to toss it in the air with a spin of their wrists. This spin is associated with centripetal acceleration, which in return leads to a *centrifugal effect* in the rotating frame that stretches the dough radially out. Inspired from this notion, we propose the creation of the centrifugal stretch deformer that reproduces such an effect with

$$\psi^{cs}(\omega, r) = -K_{cs} \, \omega \times (\omega \times r) \, . \qquad (12)$$

Note, as the centrifugal effect acts outward from the axis of rotation (as visualized in Figure 2), we negate the centripetal acceleration which is directed inward. $\psi^{cs}$ creates an effect that translates vertices outward away from the point of rotation. As the magnitude of acceleration increases linearly with distance, the vertices farther away from the center of rotation experience larger deformation than those closer to the center. A simple example appears in Figure 7.

### 5.2 Centrifugal lift

Inspired by the motion of a twirling cloth, such as a dancer's skirt, we propose a centrifugal lift deformer. As the dancer twirls, the skirt gently lifts. The same centrifugal force that caused the pizza dough to expand also applies to this shape. However, in the case of a cloth, the garment (surface) is more inextensible, and internal stress will prevent elongation in its local tangent plane. As a result, the rotating dress may first unwrap from its wrinkled configuration

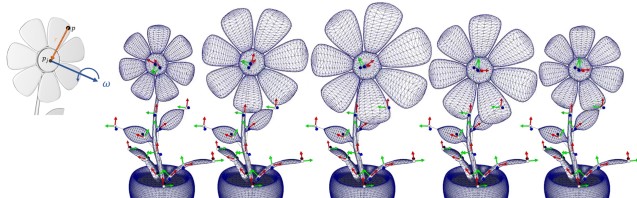

Figure 7: Centrifugal stretch effect applied onto an animation of a flower twisting its head around its central point.

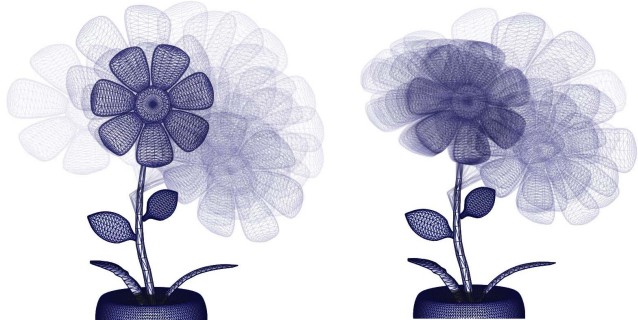

Figure 9: Comparison of untuned (left) and tuned (right) time filters. The untuned filter has frames distributed nearly equally throughout the clip. The tuned filter has more frames, with more easing at the start and end of the clip. It also has a smoother arc motion.

onto its extended shape before its remaining degree of freedom are "lifted" in the relative normal direction.

While a general solution to account for such effect is difficult without a dynamic model, we can offer a simple solution that can be used to approximate this *centrifugal lifting* effect on conical-like shapes. Cones with rotation about their center have consistent normals that allow us to filter the previous stretch deformer to act only along the normal direction and create a lift effect. Our basic formulation of this centrifugal lift $\psi^{cl}$ is thus

$$\psi^{cl}(\omega,r) = K_{cl}\,\chi(r)\,(\psi^{cs}(\omega,r)\cdot n)\,n\;, \qquad (13)$$

where $n$ is the local normal of an ideal cone at the current vertex position $p$, $K_{cl}$ is a user-defined parameter enabling to allow tuned exaggeration of the lift. $\chi(r) = \|r\| - r_0$ represents the axial co-ordinates along the cone of base radius $r_0$. This ensures that no deformation is applied at the base of the cone.

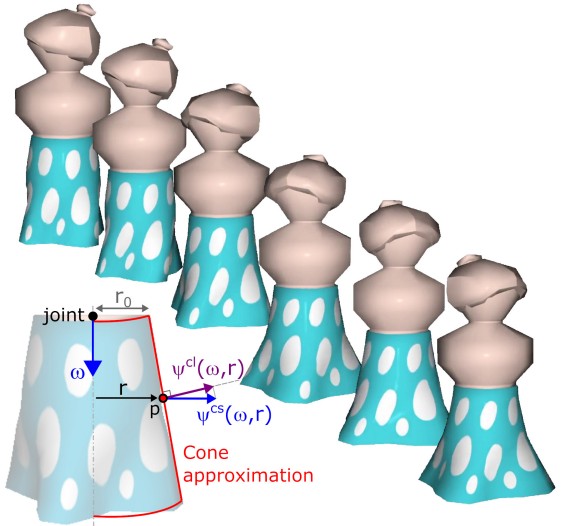

Figure 8: Centrifugal lift $\psi^{cl}$ is applied on meshes approximated by a conical surface. The deformation is constrained to act only along the normal associated to the conical approximation.

Further, any model that can be reasonably approximated by the cone can enjoy this deformer by using the cone as a proxy geometry and mapping the projected geometry to the latter. For example, the stylized dress in Figure 8 maps to the nearest point in the conical approximation to yield a set of normals that are used for the skirt deformation with $\psi^{cl}$. Note that the top part of the dress is stitched to the character thanks to the component $\chi(r)$ in Eq. (13). This effect can further be seen as a more generic deformation filter that can be applied to any other deformer $\psi$ in constraining it to act in the normal direction, and can be defined as the functional

$$\psi^{cl}_{filter} : \psi \mapsto K_{cl}\,\chi\,(\psi\cdot n)\,n\;, \qquad (14)$$

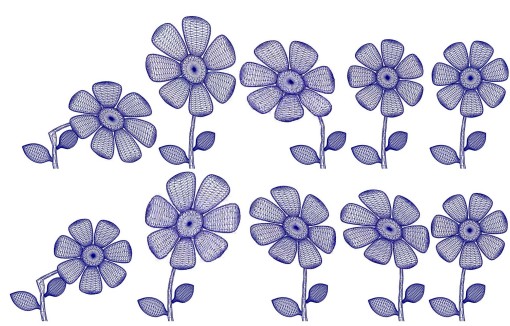

Figure 10: Comparison between the use of a small time filter (top) and a large time filter (bottom) applied on the acceleration-based drag, followthrough, and deformers. Frames (2), (3), and (5) emphasize the differences between the two filters, showing that small filters increase deformer reactivity.

where $\psi$ can be an arbitrary combination of the previously defined deformers.

## 6 TIME FILTERING

As a final extension to the AS approach, which can also be applied in the VS framework but did not appear in the previous work, we explore the use of time filtering to offer more fine grain and useful control over the mixing of various deformers. To this end, we point out that time filtering is a useful tool in general.

Time filtering brings two benefits. First, filtering inputs using a low-pass filter allows our method to handle non-smooth inputs robustly, for example, coming from direct interaction where the user articulates the skeleton using a mouse. Second, when combining effects, applying different time filters allows a controlled delay on the timing of when a specific effect will occur relative to another. We considered in our case a simple auto-regressive first order low-pass filter

$$y(t) = \gamma x(t) + (1 - \gamma)\,y(t - \Delta t)\;, \qquad (15)$$

where $y$ is the value to be filtered, $x$ represents the input (linear/angular velocity/acceleration), $\Delta t$ is the time duration, and $\gamma \in [0,1]$ is a user-defined parameter that controls the cutoff frequency. Qualitatively, a small value of $\gamma$ will lead to a smooth but delayed signal, while $\gamma$ values near one will lead to snappier response times with higher frequencies.

To apply a controlled delay to different effects, the animator sets different $\gamma$ values to each. Empirically, we found useful a

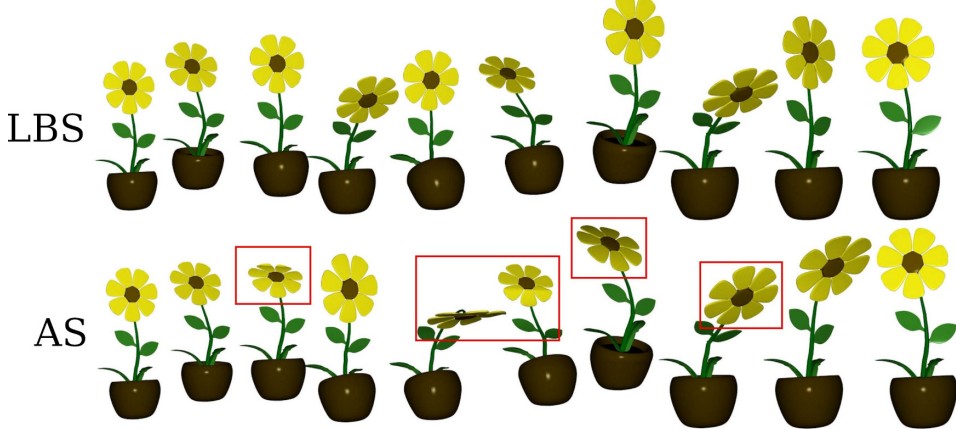

Figure 11: Flower animation with LBS (top), acceleration skinning with acceleration drag and followthrough deformation (bottom). The highlighted frames showcase the flower face bending due to acceleration skinning deformation.

long and smooth centrifugal stretch deformation ($\gamma = 0.6$), while followthrough works best as a faster and more temporary effect ($\gamma = 0.8$). This authoring of the timing of the effect is supported trivially through the described time filtering, by considering a small $\gamma$-value for $\omega$ used in $\psi^{ft}(\omega, r)$, and a larger $\gamma$-value to compute $\alpha$ used in $\psi^{cs}(\alpha, r)$. As shown in Figure 9, adjusting these values allows artistic control over the resulting animation timing on each individual effect, while preserving a simple combination of the effects as a linear summation.

## 7 RESULTS

We illustrate and compare some AS results obtained on more complex animations that can also be seen in the associated video. All the illustrations are computed in real time (approx. 60 fps) on a machine with an Intel 2.70 GHz multi-core processor and 32GB RAM, using a non-optimized CPU implementation. The cow and flower examples have a rig of with about ten joints and up to a thousand triangles. Although we did not propose a GPU implementation in this paper, the computational time of AS is similar to VS and shares the same properties: it is fully compatible with highly efficient computation in a single-pass vertex shader that was shown to be applicable up to millions of vertices [23]. In our implementation, the computational cost associated to the application of AS-deformers is roughly similar to the one spent in computing LBS. As thus, a full AS computation (LBS + AS-deformers) is twice slower than a raw LBS. However, please note that the non-sparsity of the weights $\tilde{b}$ limits the possibility of extreme optimization that can be applied on basic LBS.

Figure 11 shows the comparison of a motion made by an artist of a flower pot "jumping" from left to right. The top row shows the basic animation set by the artist using LBS, while the bottom row illustrates the result obtained after adding AS deformers. The bending of the flower obtained during the acceleration (drag effect) and deceleration phases (followthrough effect) are highlighted with the red rectangles. Figure 12 shows a comparison between VS and AS on an animated cow illustrating the conceptual example shown in Figure 3. The main difference here between AS and VS can be noticed at the end of the motion during the deceleration phase where the followthrough deformer is acting in the AS animation.

We note that an intuitive analogy can be made between the related deformers and material properties. While velocity-based deformers represent the effect of friction (or fluid drag) during a motion, acceleration-based deformers relate to momentum and inertial effects. In addition, the time-filtering models delay these effects and

can therefore be linked to modifying the strain rates of visco-elastic materials. While not explicit in their connection to these physical properties, we find these analogies useful for aiding in animators use of the described deformers.

Note, while both VS and AS propose a notion of "stretch" deformers, they differ sharply in their effect and are applicable in separate scenarios (see Figure 13). VS stretch relies on a scaling transformation to produce a squash and stretch effect. It requires an axis built from an ad-hoc frame using the skeletal structure and a notion of relative *centroid* associated to a limb and its descendant. VS stretch is relevant to illustrate cartoon effects that are elongated along the line of action and allow stretching as well as squeezing in that direction. On the contrary, AS centrifugal stretch relies on local translation only that does not require the precomputation of such *centroid*, and its direction is fully defined from the joint transformation. However, the lack of notion of *centroid* does not allow to represent directly a notion of squeezing centered around the moving limb.

## 8 CONCLUSION

We introduce Acceleration Skinning as a real-time skinning technique that employs the key components of articulated skeletal acceleration for cartoon effects. The general idea is to utilize specific acceleration terms to expand the gamut of deformation effects available through the skinning pipeline. The approach is meant to be employed in conjunction with the recent work in Velocity Skinning [23]. In this paper, we showcase its efficacy by creating three new deformation effects, including automatic followthrough, a common design tool called out by classic 2D animators. Beyond, AS is a general approach with an expandable collection of effects.

This method is not intended as a replacement to physical simulation of deformation, but rather as an artist-driven tool to create stylized secondary motion. In support of such, it is easy to control, and works "out of the box" on standard skeletal rigs. As the effects are tied to the skinning rig and not the animation, AS (and VS) are useful for real-time animation effects, even for unseen motion inputs. AS extends the capabilities of LBS and VS to empower animators through a simple tool that bootstraps existing pipelines. Further, in this paper, we explore the power of effect-specific time filtering and its relevance to enhancing artist control. However, unlike physical modeling, the technique is not able to handle collisions, interaction forces, and other nuanced dynamic effects that are generated by direct simulation. In addition, our implementation of the lift deformer is not fully generic as the conical approximation is currently

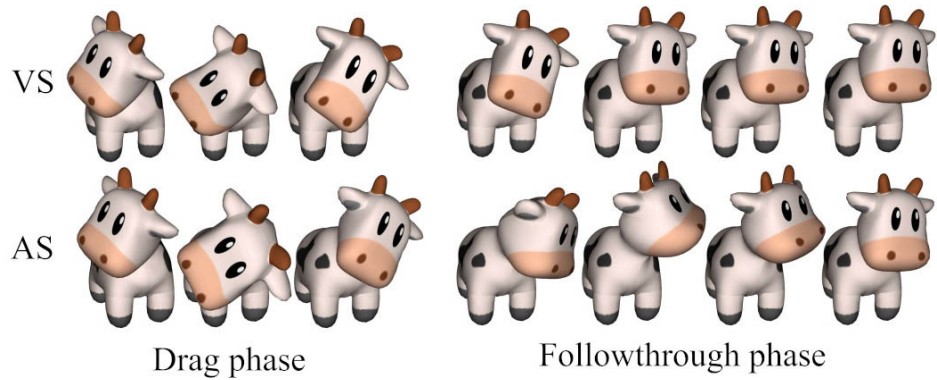

VS

AS

Drag phase

Followthrough phase

Figure 12: Comparison between velocity skinning (top) and acceleration skinning (bottom) deformation during the drag and followthrough phase.

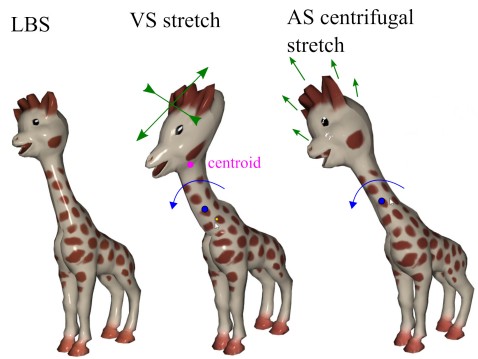

LBS      VS stretch      AS centrifugal stretch

centroid

Figure 13: Comparison between VS stretch and AS centrifugal stretch deformer effects. Green arrows indicate the direction of stretch for rotation about the blue pivot. VS stretch also requires the notion of a centroid (pink) as well as a frame that depends on the bone orientation. Both capture distinct effects. VS stretch model cartoon squash and stretch to suggest the intensity of the speed of the motion. In contrast, AS centrifugal stretch models elongation, as if due to material yield by representing the centrifugal effect of bone rotation.

pre-computed manually. Automatic conical fit could be explored in the future to apply this deformer in a seamlessly.

While we describe effects that hearken back to traditional animation, the technique is not limited to this restriction. For example, one possible future extension may be to model oscillatory motion using a skinning system. Hypothetically, if we were to expand the system to include higher order derivatives, more oscillations could be modeled. Therefore, a possible direction for future work lies in employing this technique for higher derivatives. Because the work can be applied in real-time, we are excited to investigate interactive applications, for example the use of such in gaming. As an AS rig works independently of the animation, it may be applied as an add-on for interactive character animation, such as in games or in a virtual reality (VR) setting.

In summary, we introduce the AS method that is a natural extension to VS, pushing further into the potential for deformable effects layered over LBS, and providing a wider collection of stylized deformations for artists to employ.

### ACKNOWLEDGMENTS

The authors acknowledge the foundational efforts of the original Velocity Skinning team and Bellairs Workshop on Animation 2020. We also thank Eric Patterson, Jerry Tessendorf, and the DPA program for support the first author received. Finally, thank you for the flower rig and animation courtesy of Rodney Costa and Kalya Rutherford.

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
