# OpenReview forum: "Acceleration Skinning: Kinematics-Driven Cartoon Effects for Articulated Characters"
_graphicsinterface.org/Graphics_Interface/2022/Conference — GI 2022_

### Official Review · Reviewer_ic8x · 2022-04-07
**This paper proposes acceleration skinning, which is a nice extension to velocity skinning.**

**Rating:** 9
**Confidence:** 4

**Review:**

The paper contributes to the field of realtime kinematics-based skinning techniques. It adds animation/deformation effects due to acceleration terms of the motion. It integrates nicely with LBS and VS. Time filtering is also a nice add-on component. Overall the paper is well written, and the demos are to the point.

Possible improvements:
1. the conflict between velocity drag and acceleration followthrough seems to be handled in a heuristic way. Is there any physically more principled way to deal with this? A discussion could be added here.
2. $\omega \times \omega \times r$ seems not correct. Don't you need parentheses for the last two terms?

---

### Official Review · Reviewer_AeLP · 2022-04-11
**Acceleration Skinning: Accept**

**Rating:** 9
**Confidence:** 4

**Review:**

This method proposes a new skinning technique on acceleration, an extension of of the Velocity Skinning [23]. This paper demonstrates the power of this extension to capture novel cartoon effects (e.g., follow through and centrifugal stretch).

I am leaning towards accepting this paper. I enjoy the simplicity and efficiency of the proposed solution. The results are convincing. I believe this is an important step towards capturing the fundamental principles of animation.

In general, this paper is clear and easy to follow. But I would recommend to include more background in velocity skinning to improve reproducibility. For example, I find it hard to reproduce the skinning weights computation $\tilde{b}_j$ solely based on the details in the paper. I would also recommend to include a figure showing the difference between LBS skinning weights $b_j$ and the velocity/acceleration skinning weights $\tilde{b}_j$.

In Sec 5.2, I also find the Cone Approximation a bit confusing due to the lack of details. For example, how is this cone computed? How would this method generalize to shapes that don't look like a cone (e.g., a rope, a T-shirt).

I also have a few questions related to the paper:
- I wonder why using the same skinning weights $\tilde{b}_j$ from the velocity skinning in acceleration skinning is potentially a good choice.
- I wonder whether there are relationship between the AS parameters (e.g., time filtering) and any material models so that we can e.g. use a set of AS parameters to approximate the behavior of real materials.

---

### Official Review · Reviewer_5YWX · 2022-04-13
**Acceleration Deformers for LBS animation**

**Rating:** 6
**Confidence:** 3

**Review:**

This paper extends the method of velocity skinning to also take into account acceleration. Three new deformers which use the acceleration are presented. The approach works with linear blend skinning (LBS), but would as well work with dual quaternions. A filtering strategy is also presented.

The paper is quite well written and structured with only few typos.

The method feels a bit ad hoc. Nevertheless, even though the paper does not present something ground breaking, the proposed approach still seems interesting to reproduce the typical drag and followthrough effects from cartoon-style animation. Also, the method is simple to understand and implement (except for Sec. 5.2).

Some effects of the deformers remain a bit of a mystery to me. An intuitive explanation would help the reader. In Fig. 5, the petals stretch (most evident in the 4th image from the left) while in Fig. 6, the petals shrink (most evident in the 2nd image from the left). It is not clear which aspect of the deformer causes this scaling of the petals.

The centrifugal lift effect (Sec. 5.2) would be hard to reproduce given the current explanations. Is the cone recomputed every frame? Or is it that there is a different cone at every point p? In Fig. 8, it seems that the points on the “skirt” which are toward the top do not move, while the ones toward the bottom move a lot. How can that be achieved with a cone (which has the same side normal from top to bottom)?

Still for the centrifugal lift effect, I do not see how the “skirt” will get back to its original shape.

Sec. 6: The explanation with the non-smooth inputs from direct interaction is misleading. I had to read this twice to convince me that, in the end, there are no such non-smooth inputs and the filtering is used on the smooth inputs.

The type of CPU / GPU should be reported. If someone reads the paper in 4, 8 or 12 years, it will be hard to figure this out from “common laptop”.

It would be useful to report the fraction of time required for LPS + AS as well as for just AS, compared to the whole computation (animation, rendering, LBS, AS, etc.).

“Traditional skinning defines (static) skin poses”: Seems strange. Shouldn’t it be “Traditional keyframe animation defines (static) skeleton poses?”

Fig. 2: Should $\alpha_j$ be lined up with $\omega_j$?

I would use “I” to denote the identity matrix instead of “Id”

Typos and such:
* “Rohmer et al [23]”: missing “.” for “et al.”
* Missing parenthesis in “(see Figure 3.”
* “allowing the scale of the magnitude of each deformation” scaling?
* “We take employ”
* Sec. 5, missing space in “centrifugal “lift” deformer,$\psi^{cl}$ ”
* Missing space “visualized in Figure2”

---

### Decision · Program_Chairs · 2022-04-17

Accept